# Efficient Multi-objective Neural Architecture Search via Lamarckian Evolution

**Thomas Elsken**
Bosch Center for Artificial Intelligence
and University of Freiburg
Thomas.Elsken@de.bosch.com

**Jan Hendrik Metzen**
Bosch Center for Artificial Intelligence
JanHendrik.Metzen@de.bosch.com

**Frank Hutter**
University of Freiburg
fh@cs.uni-freiburg.de

## Abstract

Neural Architecture Search aims at automatically finding neural network architectures that are competitive with architectures designed by human experts. While recent approaches have achieved state-of-the-art predictive performance for, e.g., image recognition, they are problematic under resource constraints for two reasons: (1) the neural architectures found are solely optimized for high predictive performance, without penalizing excessive resource consumption; (2) most architecture search methods require vast computational resources. We address the first shortcoming by proposing LEMONADE, an evolutionary algorithm for multi-objective architecture search that allows approximating the entire Pareto front of architectures under multiple objectives, such as predictive performance and number of parameters, in a single run of the method. We address the second shortcoming by proposing a Lamarckian inheritance mechanism for LEMONADE which generates child networks that are warm started with the predictive performance of their trained parents. This is accomplished by using (approximate) network morphism operators for generating children. The combination of these two contributions allows finding models that are on par or even outperform both hand-crafted as well as automatically-designed networks.

## 1 Introduction

Deep learning has enabled remarkable progress on a variety of perceptual tasks, such as image recognition (Krizhevsky et al., 2012), speech recognition (Hinton et al., 2012), and machine translation (Bahdanau et al., 2015). One crucial aspect for this progress are novel *neural architectures* (Szegedy et al., 2016; He et al., 2016; Huang et al., 2017b). Currently employed architectures have mostly been developed manually by human experts, which is a time-consuming and error-prone process. Because of this, there is growing interest in automatic *architecture search* methods (Elsken et al., 2018). Some of the architectures found in an automated way have already outperformed the best manually-designed ones; however, algorithms such as by Zoph & Le (2017); Zoph et al. (2018); Real et al. (2017; 2018) for finding these architectures require enormous computational resources often in the range of thousands of GPU days.

Prior work on architecture search has typically framed the problem as a single-objective optimization problem. However, most applications of deep learning do not only require high predictive performance on unseen data but also low *resource-consumption* in terms of, e.g., inference time, model size or energy consumption. Moreover, there is typically an implicit trade-off between predictive performance and consumption of resources. Recently, several architectures have been manually designed that aim at reducing resource-consumption while retaining high predictive performance (Iandola et al., 2016; Howard et al., 2017; Sandler et al., 2018). Automatically found neural architectures have also been down-scaled to reduce resource consumption (Zoph et al., 2018). However, very little previous work has taken the trade-off between resource-consumption and predictive performance into account during automatic architecture search.

In this work, we make the following two main contributions:

1. To overcome the need for thousands of GPU days (Zoph & Le, 2017; Zoph et al., 2018; Real et al., 2018), we make use of operators acting on the space of neural network architectures that preserve the function a network represents, dubbed *network morphisms* (Chen et al., 2015; Wei et al., 2016), obviating training from scratch and thereby substantially reducing the required training time per network. This mechanism can be interpreted as Lamarckian inheritance in the context of evolutionary algorithms, where Lamarckism refers to a mechanism which allows passing skills acquired during an individual's lifetime (e.g., by means of learning), on to children by means of inheritance. Since network morphisms are limited to solely increasing a network's size (and therefore likely also resource consumption), we introduce *approximate network morphisms* (Section 3.2) to also allow shrinking networks, which is essential in the context of multi-objective search. The proposed Lamarckian inheritance mechanism could in principle be combined with any evolutionary algorithm for architecture search, or any other method using (a combination of) localized changes in architecture space.

2. We propose a **L**amarckian **E**volutionary algorithm for **M**ulti-**O**bjective **N**eural **A**rchitecture **DE**sign, dubbed LEMONADE, Section 4, which is suited for the joint optimization of several objectives, such as predictive performance, inference time, or number of parameters. LEMONADE maintains a population of networks on an approximation of the Pareto front of the multiple objectives. In contrast to generic multi-objective algorithms, LEMONADE exploits that evaluating certain objectives (such as an architecture's number of parameters) is cheap while evaluating the predictive performance on validation data is expensive (since it requires training the model first). Thus, LEMONADE handles its various objectives differently: it first selects a subset of architectures, assigning higher probability to architectures that would fill gaps on the Pareto front for the "cheap" objectives; then, it trains and evaluates only this subset, further reducing the computational resource requirements during architecture search. In contrast to other multi-objective architecture search methods, LEMONADE (i) does not require to define a trade-off between performance and other objectives a-priori (e.g., by weighting objectives when using scalarization methods) but rather returns a set of architectures, which allows the user to select a suitable model a-posteriori; (ii) LEMONADE does not require to be initialized with well performing architectures; it can be initialized with trivial architectures and hence requires less prior knowledge. Also, LEMONADE can handle various search spaces, including complex topologies with multiple branches and skip connections.

We evaluate LEMONADE for up to five objectives on two different search spaces for image classification: (i) non-modularized architectures and (ii) cells that are used as repeatable building blocks within an architecture (Zoph et al., 2018; Zhong et al., 2018) and also allow transfer to other data sets. LEMONADE returns a population of CNNs covering architectures with 10 000 to 10 000 000 parameters.

Within only 5 days on 16 GPUs, LEMONADE discovers architectures that are competitive in terms of predictive performance and resource consumption with hand-designed networks, such as MobileNet V2 (Sandler et al., 2018), as well as architectures that were automatically designed using 40x greater resources (Zoph et al., 2018) and other multi-objective methods (Dong et al., 2018).

## 2 BACKGROUND AND RELATED WORK

**Multi-objective Optimization** Multi-objective optimization (Miettinen, 1999) deals with problems that have multiple, complementary objective functions $f_1, \ldots, f_n$. Let $\mathcal{N}$ be the space of feasible solutions $N$ (in our case the space of feasible neural architectures). In general, multi-objective optimization deals with finding $N^* \in \mathcal{N}$ that minimizes the objectives $f_1, \ldots, f_n$. However, typically there is no single $N^*$ that minimizes all objectives at the same time. In contrast, there are multiple *Pareto-optimal* solutions that are optimal in the sense that one cannot reduce any $f_i$ without increasing at least one $f_j$. More formally, a solution $N^{(1)}$ Pareto-dominates another solution $N^{(2)}$ if $\forall i \in 1, \ldots, n : f_i(N^{(1)}) \leq f_i(N^{(2)})$ and $\exists j \in 1, \ldots, n : f_j(N^{(1)}) < f_j(N^{(2)})$. The Pareto-optimal solutions $N^*$ are exactly those solutions that are not dominated by any other $N \in \mathcal{N}$. The set of Pareto optimal $N^*$ is the so-called *Pareto front*.

**Neural Architecture Search** We give a short overview of the most relevant works in neural architecture search (NAS) and refer to Elsken et al. (2018) for a comprehensive survey on the topic. It

was recently proposed to frame NAS as a *reinforcement learning* (RL) problem, where the reward of the RL agent is based on the validation performance of the trained architecture (Baker et al., 2017a; Zoph & Le, 2017; Zhong et al., 2018; Pham et al., 2018). Zoph & Le (2017) use a recurrent neural network to generate a string representing the neural architecture. In a follow-up work, Zoph et al. (2018) search for *cells*, which are repeated according to a fixed macro architecture to generate the eventual architecture. Defining the architecture based on a cell simplifies the search space.

An alternative to using RL are *neuro-evolutionary* approaches that use genetic algorithms for optimizing the neural architecture (Stanley & Miikkulainen, 2002; Liu et al., 2018a; Real et al., 2018; Miikkulainen et al., 2017; Xie & Yuille, 2017). In contrast to these works, our proposed method is applicable for multi-objective optimization and employs Lamarckian inheritance, i.e, learned parameters are passed on to a network's offspring. A related approach to our Lamarckian evolution is population-based training (Jaderberg et al., 2017), which, however, focuses on hyperparameter optimization and not on the specific properties of the optimization of neural architectures. We note that it would be possible to also include the evolution of hyperparameters in our work.

Unfortunately, most of the aforementioned approaches require vast computational resources since they need to train and validate thousands of neural architectures; e.g., Zoph & Le (2017) trained over 10.000 neural architectures, requiring thousands of GPU days. One way of speeding up evaluation is to *predict performance* of a (partially) trained model (Domhan et al., 2015; Baker et al., 2017b; Klein et al., 2017; Liu et al., 2017). Works on performance prediction are complementary to our work and could be incorporated in the future.

One-Shot Architecture Search is another promising approach for speeding up performance estimation, which treats all architectures as different subgraphs of a supergraph (the one-shot model) and shares weights between architectures (Saxena & Verbeek, 2016; Brock et al., 2017; Pham et al., 2018; Liu et al., 2018b; Bender et al., 2018). Only the weights of a single one-shot model need to be trained, and architectures (which are just subgraphs of the one-shot model) can then be evaluated without any separate training. However, a general limitation of one-shot NAS is that the supergraph defined a-priori restricts the search space to its subgraphs. Moreover, approaches which require that the entire supergraph resides in GPU memory during architecture search will be restricted to relatively small supergraphs. It is also not obvious how one-shot models could be employed for multi-objective optimization as all subgraphs of the one-shot models are of roughly the same size and it is not clear if weight sharing would work for very different-sized architectures. LEMONADE does not suffer from any of these disadvantages; it can handle arbitrary large, unconstrained search spaces while still being efficient.

Elsken et al. (2017); Cai et al. (2018a) proposed to employ the concept of *network morphisms* (see Section 3.1). The basic idea is to initialize weights of newly generated neural architectures based on weights of similar, already trained architectures so that they have the same accuracy. This pretrained initialization allows reducing the large cost of training all architectures from scratch. Our work extends this approach by introducing *approximate network morphisms*, making the use of such operators suitable for multi-objective optimization.

**Multi-objective Neural Architecture Search**   Very recently, there has also been some work on multi-objective neural architecture search (Kim et al., 2017; Dong et al., 2018; Tan et al., 2018) with the goal of not solely optimizing the accuracy on a given task but also considering resource consumption. Kim et al. (2017) parameterize an architecture by a fixed-length vector description, which limits the architecture search space drastically. In parallel, independent work to ours, Dong et al. (2018) extend PNAS (Liu et al., 2017) by considering multiple objective during the model selection step. However, they employ CondenseNet (Huang et al., 2017a) as a base network and solely optimize building blocks within the network which makes the search less interesting as (i) the base network is by default already well performing and (ii) the search space is again limited. Tan et al. (2018) use a weighted product method (Deb & Kalyanmoy, 2001) to obtain a scalarized objective. However, this scalarization comes with the drawback of weighting the objectives *a-priori*, which might not be suitable for certain applications. In contrast to all mentioned work, LEMONADE (i) does not require a complex macro architecture but rather can start from trivial initial networks, (ii) can handle arbitrary search spaces, (iii) does not require to define hard constraints or weights on objectives a-priori.

## 3 NETWORK OPERATORS

Let $\mathcal{N}(\mathcal{X})$ denote a space of neural networks, where each element $N \in \mathcal{N}(\mathcal{X})$ is a mapping from $\mathcal{X} \subset \mathbb{R}^n$ to some other space, e.g., mapping images to labels. A network operator $T : \mathcal{N}(\mathcal{X}) \times \mathbb{R}^k \to \mathcal{N}(\mathcal{X}) \times \mathbb{R}^j$ maps a neural network $N^w \in \mathcal{N}(\mathcal{X})$ with parameters $w \in \mathbb{R}^k$ to another neural network $(TN)^{\tilde{w}} \in \mathcal{N}(\mathcal{X}), \tilde{w} \in \mathbb{R}^j$.

We now discuss two specific classes of network operators, namely *network morphisms* and *approximate network morphisms*. Operators from these two classes will later on serve as mutations in our evolutionary algorithm.

### 3.1 NETWORK MORPHISMS

Chen et al. (2015) introduced two function-preserving operators for deepening and widening a neural network. Wei et al. (2016) built upon this work, dubbing function-preserving operators on neural networks *network morphisms*. Formally, a network morphism is a network operator satisfying $N^w(x) = (TN)^{\tilde{w}}(x)$ for every $x \in \mathcal{X}$, i.e., $N^w$ and $(TN)^{\tilde{w}}$ represent the same function. This can be achieved by properly initializing $\tilde{w}$.

We now describe the operators used in `LEMONADE` and how they can be formulated as a network morphism. We refer to Appendix A.1.1 for details.

1. Inserting a Conv-BatchNorm-ReLU block. We initialize the convolution to be an identity mapping, as done by Chen et al. (2015) ("Net2DeeperNet"). Offset and scale of BatchNormalization are initialized to be the (moving) batch mean and (moving) batch variance, hence initially again an identity mapping. Since the ReLU activation is idempotent, i.e., $ReLU(ReLU(x)) = ReLU(x)$, we can add it on top of the previous two operations without any further changes, assuming that the block will be added on top of a ReLU layer.

2. Increase the number of filters of a convolution. This operator requires the layer to be changed to have a subsequent convolutional layer, whose parameters are padded with 0's. Alternatively, one could use the "Net2WiderNet" operator by Chen et al. (2015).

3. Add a skip connection. We allow skip connection either by concatenation (Huang et al., 2017b) or by addition (He et al., 2016). In the former case, we again use zero-padding in sub-sequential convolutional layers. In the latter case, we do not simply add two outputs $x$ and $y$ but rather use a convex combination $(1 - \lambda)x + \lambda y$, with a learnable parameter $\lambda$ initialized as 0 (assuming $x$ is the original output and $y$ the output of an earlier layer).

### 3.2 APPROXIMATE NETWORK MORPHISMS

One common property of all network morphisms is that they can only increase the capacity of a network[1]. This may be a reasonable property if one solely aims at finding a neural architectures with maximal accuracy, but not if one also aims at neural architectures with low resource requirements. Also, decisions once made can not be reverted. Operators like removing a layer could considerably decrease the resources required by the model while (potentially) preserving its performance.

Hence, we now generalize the concept of network morphisms to also cover operators that reduce the capacity of a neural architecture. We say an operator $T$ is an approximate network morphism (ANM) with respect to a neural network $N^w$ with parameters $w$ if $N^w(x) \approx (TN)^{\tilde{w}}(x)$ for every $x \in \mathcal{X}$. We refer to Appendix A.1.2 for a formal definition. In practice we simply determine $\tilde{w}$ so that $\tilde{N}$ approximates $N$ by using knowledge distillation (Hinton et al., 2015).

In our experiments, we employ the following ANM's: (i) remove a randomly chosen layer or a skip connection, (ii) prune a randomly chosen convolutional layer (i.e., remove $1/2$ or $1/4$ of its filters), and (iii) substitute a randomly chosen convolution by a depthwise separable convolution. Note that these operators could easily be extended by sophisticated methods for compressing neural networks (Han et al., 2016; Cheng et al., 2018).

---

[1]If one would decrease the network's capacity, the function-preserving property could in general not be guaranteed.

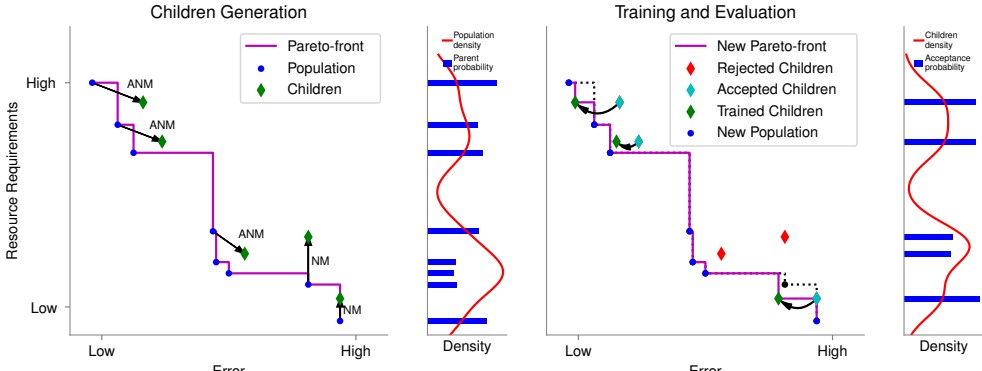

Figure 1: Conceptual illustration of LEMONADE. (Left) LEMONADE maintains a population of trained networks that constitute a Pareto front in the multi-objective space. Parents are selected from the population inversely proportional to their density. Children are generated by mutation operators with Lamarckian inheritance that are realized by network morphisms and approximate network morphisms. NM operators generate children with the same initial error as their parent. In contrast, children generated with ANM operators may incur a (small) increase in error compared to their parent. However, their initial error is typically still very small. (Right) Only a subset of the generated children is accepted for training. After training, the performance of the children is evaluated and the population is updated to be the Pareto front.

## 4 LEMONADE: MULTI-OBJECTIVE NEURAL ARCHITECTURE SEARCH

In this section, we propose a **L**amarckian **E**volutionary algorithm for **M**ulti-**O**bjective **N**eural **A**rchitecture **DE**sign, dubbed LEMONADE. We refer to Figure 1 for an illustration as well as Algorithm 1 for pseudo code. LEMONADE aims at minimizing multiple objectives $\mathfrak{f}(N) = (f_{exp}(N), f_{cheap}(N))^\top \in \mathbb{R}^m \times \mathbb{R}^n$, whose first components $f_{exp}(N) \in \mathbb{R}^m$ denote expensive-to-evaluate objectives (such as the validation error or some measure only be obtainable by expensive simulation) and its other components $f_{cheap}(N) \in \mathbb{R}^n$ denote cheap-to-evaluate objectives (such as model size) that one also tries to minimize. LEMONADE maintains a population $\mathcal{P}$ of parent

---

**Algorithm 1** LEMONADE

1: input: $\mathcal{P}_0, \mathfrak{f}, n_{gen}, n_{pc}, n_{ac}$
2: $\mathcal{P} \leftarrow \mathcal{P}_0$
3: **for** $i \leftarrow 1, \ldots, n_{gen}$ **do**
4:     $p_{KDE} \leftarrow KDE\big(\{f_{cheap}(N)|N \in \mathcal{P}\}\big)$
5:     Compute parent distribution $p_\mathcal{P}$ (Eq. 1)
6:     $\mathbf{N}^c_{pc} \leftarrow GenerateChildren(\mathcal{P}, p_\mathcal{P}, n_{pc})$
7:     Compute children distribution $p_{child}$ (Eq. 2)
8:     $\mathbf{N}^c_{ac} \leftarrow AcceptSubSet(\mathbf{N}^c_{pc}, p_{child}, n_{ac})$
9:     Evaluate $f_{exp}$ for $N^c \in \mathbf{N}^c_{ac}$
10:     $\mathcal{P} \leftarrow ParetoFront(\mathcal{P} \cup \mathbf{N}^c_{ac}, \mathfrak{f})$
11: **end for**
12: **return** $\mathcal{P}$

---

networks, which we choose to comprise all non-dominated networks with respect to $\mathfrak{f}$, i.e., the current approximation of the Pareto front[2]. In every iteration of LEMONADE, we first sample parent networks with respect to some probability distribution based on the cheap objectives and generate child networks by applying network operators (described in Section 3). In a second sampling stage, we sample a subset of children, again based on cheap objectives, and solely this subset is evaluated on the expensive objectives. Hence, we exploit that $f_{cheap}$ is cheap to evaluate in order to bias both sampling processes towards areas of $f_{cheap}$ that are sparsely populated. We thereby evaluate $f_{cheap}$ many times in order to end up with a diverse set of children in sparsely populated regions of the objective space, but evaluate $f_{exp}$ only a few times.

More specifically, LEMONADE first computes a density estimator $p_{KDE}$ (e.g., in our case, a kernel density estimator) on the cheap objective values of the current population, $\{f_{cheap}(N)|N \in \mathcal{P}\}$. Note that we explicitly only compute the KDE with respect to $f_{cheap}$ rather than $\mathfrak{f}$ as this allows

---

[2]One could also include some dominated architectures in the population to increase diversity, but we do not consider this in this work.

to evaluate $p_{KDE}(f_{cheap}(N))$ very quickly. Then, larger number $n_{pc}$ of *proposed children* $\mathbf{N}^c_{pc} = \{N^c_1, \ldots, N^c_{n_{pc}}\}$ is generated by applying network operators, where the parent $N$ for each child is sampled according to a distribution inversely proportional to $p_{KDE}$,

$$p_{\mathcal{P}}(N) = \frac{c}{p_{KDE}(f_{cheap}(N))}, \tag{1}$$

with a normalization constant $c = \big( \sum_{N \in \mathcal{P}} 1/p_{KDE}(f_{cheap}(N)) \big)^{-1}$. Since children have similar objective values as their parents (network morphisms do not change architectures drastically), this sampling distribution of the parents is more likely to also generate children in less dense regions of $f_{cheap}$. Afterwards, we again employ $p_{KDE}$ to sample a subset of $n_{ac}$ *accepted children* $\mathbf{N}^c_{ac} \subset \mathbf{N}^c_{pc}$. The probability of a child being accepted is

$$p_{child}(N^c) = \frac{\hat{c}}{p_{KDE}(f_{cheap}(N^c))}, \tag{2}$$

with $\hat{c}$ being another normalization constant. Only these accepted children are evaluated according to $f_{exp}$. By this two-staged sampling strategy we generate and evaluate more children that have the potential to fill gaps in $\mathfrak{f}$. We refer to the ablation study in Appendix A.2.2 for an empirical comparison of this sampling strategy to uniform sampling. Finally, LEMONADE computes the Pareto front from the current generation and the generated children, yielding the next generation. The described procedure is repeated for a prespecified number of generations (100 in our experiments).

## 5 EXPERIMENTS

We present results for LEMONADE on searching neural architectures for CIFAR-10. We ran LEMONADE with three different settings: (i) we optimize 5 objectives and search for entire architectures (Section 5.1), (ii) we optimize 2 objectives and search for entire architectures (Appendix A.2), and (iii) we optimize 2 objectives and search for cells (Section 5.2, Appendix A.2). We also transfer the discovered cells from the last setting to ImageNet (Section 5.4) and its down-scaled version ImageNet64x64 (Chrabaszcz et al., 2017) (Section 5.3). All experimental details, such as a description of the search spaces and hyperparameters can be found in Appendix A.3.

The progress of LEMONADE for setting (ii) is visualized in Figure 2. The Pareto front improves over time, reducing the validation error while covering a wide regime of, e.g., model parameters, ranging from 10 000 to 10 000 000.

### 5.1 EXPERIMENTS ON CIFAR-10

We aim at solving the following multi-objective problem: minimize the five objectives (i) performance on CIFAR-10 (expensive objective), (ii) performance on CIFAR-100 (expensive), (iii) number of parameters (cheap), (iv) number of multiply-add operations (cheap) and (v) inference time [3] (cheap). We think having five objectives is a realistic scenario for most NAS applications. Note that one could easily use other, more sophisticated measures for resource efficiency.

In this experiment, we search for entire neural network architectures (denoted as Search Space I, see Appendix A.3.2 for details) instead of convolutional cells (which we will do in a later experiment). LEMONADE natively handles this unconstrained, arbitrarily large search space, whereas other methods are by design a-priori restricted to relatively small search spaces (Bender et al., 2018; Liu et al., 2018b). Also, LEMONADE is initialized with trivial architectures (see Appendix A.3.2) rather than networks that already yield state-of-the-art performance (Cai et al., 2018b; Dong et al., 2018). The set of operators to generate child networks we consider in our experiments are the three network morphism operators (insert convolution, insert skip connection, increase number of filters), as well as the three approximate network morphism operators (remove layer, prune filters, replace layer) described in Section 3. The operators are sampled uniformly at random to generate children. The experiment ran for approximately 5 days on 16 GPUs in parallel. The resulting Pareto front consists of approximately 300 neural network architectures.

---

[3]I.e., the time required for a forward pass through the network. In detail, we measured the time for doing inference on a batch of 100 images on a Titan X GPU. We averaged over 5,000 forward passes to obtain low-variance estimates.

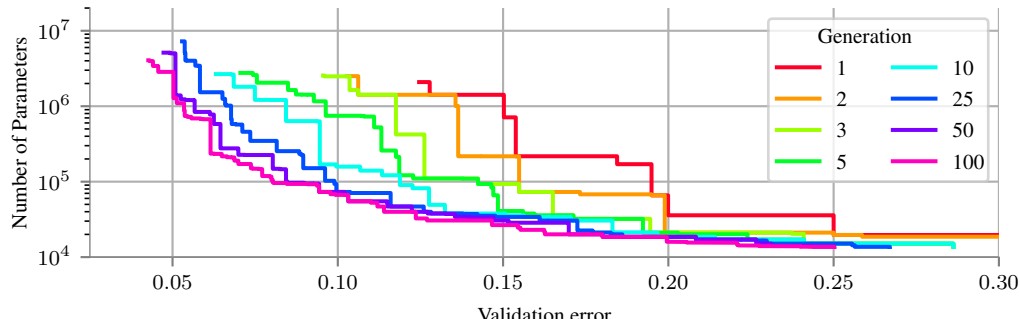

Figure 2: Progress of the Pareto front of LEMONADE during architecture search. The Pareto front gets more and more densely settled over the course of time. Very large models found (e.g., in generation 25) are discarded in a later generation as smaller, better ones are discovered. Note: generation 1 denotes the generation after one iteration of LEMONADE.

We compare against different-sized NASNets (Zoph et al., 2018) and MobileNets V2 (Sandler et al., 2018). In order to ensure that differences in test error are actually caused by differences in the discovered architectures rather than different training conditions, we retrained all architectures from scratch using exactly the same optimization pipeline with the same hyperparameters. We do not use stochastic regularization techniques, such as Shake-Shake (Gastaldi, 2017) or ScheduledDropPath (Zoph et al., 2018) in this experiment as they are not applicable to all networks out of the box.

The results are visualized in Figure 3. As one would expect, the performance on CIFAR-10 and CIFAR-100 is highly correlated, hence the resulting Pareto fronts only consist of a few elements and differences are rather small (top left). When considering the performance on CIFAR-10 versus the number of parameters (top right) or multiply-add operations (bottom left), LEMONADE is on par with NASNets and MobileNets V2 for resource-intensive models while it outperforms them in the area of very efficient models (e.g., less than 100,000 parameters). In terms of inference time (bottom right), LEMONADE clearly finds models superior to the baselines. We highlight that this result has been achieved based on using only 80 GPU days for LEMONADE compared to 2000 in Zoph et al. (2018) and with a significantly more complex Search Space I (since the entire architecture was optimized and not only a convolutional cell).

We refer to Appendix A.2 for an experiment with additional baselines (e.g., random search) and an ablation study.

## 5.2 COMPARISON TO PUBLISHED RESULTS ON CIFAR-10.

Above, we compared different models when trained with the exact same data augmentation and training pipeline. We now also briefly compare LEMONADE's performance to results reported in the literature. We apply two widely used methods to improve results over the training pipeline used above: (i) instead of searching for entire architectures, we search for cells that are employed within a hand-crafted macro architecture, meaning one replaces repeating building blocks in the architecture with discovered cells (Cai et al., 2018b; Dong et al., 2018) and (ii) using stochastic regularization techniques, such as Scheduled-DropPath during training (Zoph et al., 2018; Pham et al., 2018; Cai et al., 2018b). In our case, we run LEMONADE to search for cells within the Shake-Shake macro architecture (i.e., we replace basic convolutional blocks with cells) and also use Shake-Shake regularization (Gastaldi, 2017).

| Method | Params | Error (%) |
|---|---|---|
| DPP-Net | 0.5M | 4.62 |
| LEMONADE | 0.5M | 4.57 |
| DPP-Net | 1.0M | 4.78 |
| LEMONADE | 1.1M | 3.69 |
| NASNet | 3.3M | 2.65 |
| ENAS | 4.6M | 2.89 |
| PLNT | 5.7M | 2.49 |
| LEMONADE | 4.7M | 3.05 |
| DPP-Net | 11.4M | 4.36 |
| PLNT | 14.3M | 2.30 |
| LEMONADE | 13.1M | 2.58 |

Table 1: Comparison of LEMONADE with other NAS methods on CIFAR-10 for different-sized models under identical training conditions.

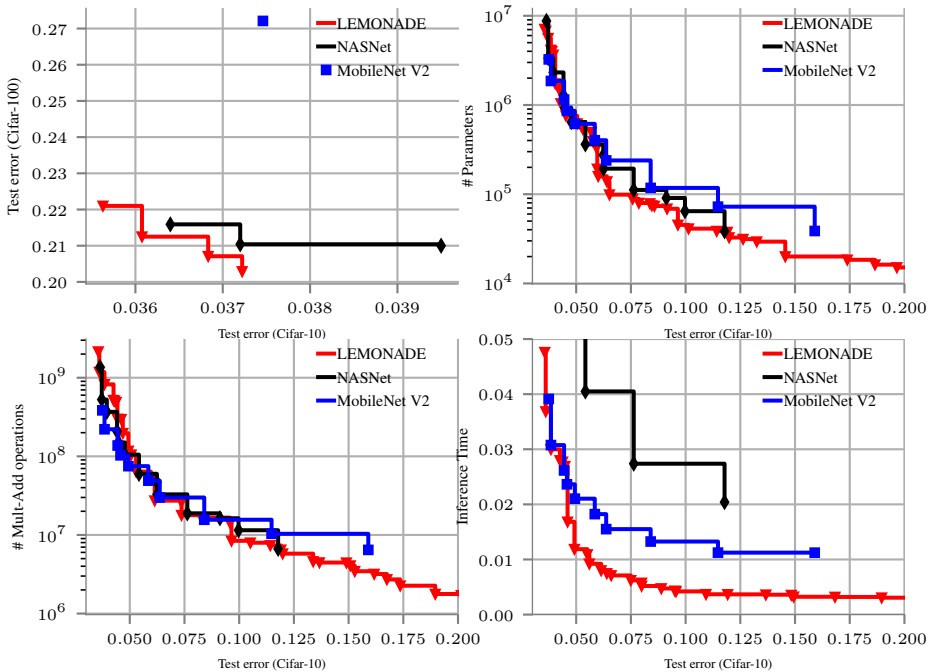

Figure 3: Comparison of LEMONADE with NASNet and MobileNet V2. LEMONADE optimized five objectives: performance on CIFAR-10 (x-axis in all plots), performance on CIFAR-100 (top left), number of parameters (top right), number of multiply add operations (bottom left) and inference time (bottom right, measured in seconds on a Titan X GPU).

We compare LEMONADE to the state of the art single-objective methods by Zoph et al. (2018) (NAS-Net), Pham et al. (2018) (ENAS) and Cai et al. (2018b) (PLNT), as well as with the multi-objective method by Dong et al. (2018) (DPP-Net). The results are summarized in Table 1. LEMONADE is on par or outperforms DPP-Net across all parameter regimes. As all other methods solely optimize for accuracy, they do not evaluate models with few parameters. However, also for larger models, LEMONADE is competitive to methods that require significantly more computational resources (Zoph et al., 2018) or start their search with non-trivial architectures (Cai et al., 2018b; Dong et al., 2018).

## 5.3 TRANSFER TO IMAGENET64x64

To study the transferability of the discovered cells to a different dataset (without having to run architecture search itself on the target dataset), we built architectures suited for ImageNet64x64 (Chrabaszcz et al., 2017) based on five cells discovered on CIFAR-10. We vary (1) the number of cells per block and (2) the number of filters in the last block to obtain different architectures for a single cell (as done by Zoph et al. (2018) for NASNets). We compare against different sized MobileNets V2, NASNets and Wide Residual Networks (WRNs) (Zagoruyko & Komodakis, 2016). For direct comparability, we again train all architectures in the same way.

In Figure 4, we plot the Pareto Front from all cells combined, as well as the Pareto Front from a single cell, Cell 2, against the baselines. Both clearly dominate NASNets, WRNs and MobileNets V2 over the entire parameter range, showing that a multi-objective search again is beneficial.

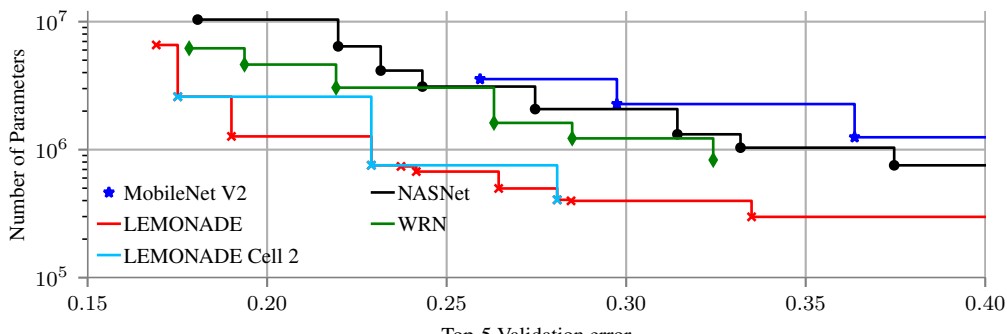

Figure 4: Transferring the cells discovered on CIFAR-10 to ImageNet64x64. A single Cell, namely Cell 2, outperforms all baselines. Utilizing 5 different cells (red line) further improves the results.

## 5.4 Transfer to ImageNet (mobile setting)

We also evaluated one discovered cell, Cell 2, on the regular ImageNet benchmark for the "mobile setting" (i.e., networks with $4M$ to $6M$ parameters and less than $600M$ multiply-add operations). The cell found by LEMONADE achieved a top-1 error of $28.3\%$ and a top-5 error of $9.6\%$; this is slightly worse than published results for, e.g., NASNet ($26\%$ and $8.4\%$, respectively) but still competitive, especially seeing that (due to time and resource constraints), we used an off-the-shelf training pipeline, on a single GPU (for four weeks), and did not alter any hyperparameters. We believe that our cell could perform substantially better with a better optimization pipeline and properly tuned hyperparameters (as in many other NAS papers by authors with more compute resources).

## 6 Conclusion

We have proposed LEMONADE, a multi-objective evolutionary algorithm for architecture search. The algorithm employs a Lamarckian inheritance mechanism based on (approximate) network morphism operators to speed up the training of novel architectures. Moreover, LEMONADE exploits the fact that evaluating several objectives, such as the performance of a neural network, is orders of magnitude more expensive than evaluating, e.g., a model's number of parameters. Experiments on CIFAR-10 and ImageNet64x64 show that LEMONADE is able to find competitive models and cells both in terms of accuracy and of resource efficiency.

We believe that using more sophisticated concepts from the multi-objective evolutionary algorithms literature and using other network operators (e.g., crossovers and advanced compression methods) could further improve LEMONADE's performance in the future.

**Acknowledgements**   We would like to thank Arber Zela and the anonymous reviewers for valuable feedback on this work. We would like to thank Nicole Finnie for supporting us with the ImageNet experiments. This work has partly been supported by the European Research Council (ERC) under the European Union's Horizon 2020 research and innovation programme under grant no. 716721.

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

# A APPENDIX

## A.1 DETAILS ON NETWORK OPERATORS

In the following two subsections we give some detailed information on the network morphisms and approximate network morphisms employed in our work.

### A.1.1 DETAILS ON NETWORK MORPHISMS

A network morphism is a network operator satisfying the network morphism equation:

$$N^w(x) = (TN)^{\tilde{w}}(x) \text{ for every } x \in \mathcal{X}, \tag{3}$$

i.e., $N^w$ and $(TN)^{\tilde{w}}$ represent the same function. This can be achieved by properly initializing $\tilde{w}$.

**Network morphism Type I.** Let $N_i^{w_i}(x)$ be some part of a neural architecture $N^w(x)$, e.g., a layer or a subnetwork. We replace $N_i^{w_i}$ by

$$\tilde{N}_i^{\tilde{w}_i}(x) = AN_i^{w_i}(x) + b, \tag{4}$$

with $\tilde{w}_i = (w_i, A, b)$. The network morphism equation (3) then holds for $A = \mathbf{1}, b = \mathbf{0}$. This morphism can be used to add a fully-connected or convolutional layer, as these layers are simply linear mappings. Chen et al. (2015) dubbed this morphism "Net2DeeperNet". Alternatively to the above replacement, one could also choose

$$\tilde{N}_i^{\tilde{w}_i}(x) = C(AN_i^{w_i}(x) + b) + d, \tag{5}$$

with $\tilde{w}_i = (w_i, C, d)$. $A, b$ are fixed, non-learnable. In this case, network morphism Equation (3) holds if $C = A^{-1}, d = -Cb$. A Batch Normalization layer (or other normalization layers) can be written in the above form: $A, b$ represent the batch statistics and $C, d$ the learnable scaling and shifting.

**Network morphism Type II.** Assume $N_i^{w_i}$ has the form $N_i^{w_i}(x) = Ah^{w_h}(x) + b$ for an arbitrary function $h$. We replace $N_i^{w_i}$, $w_i = (w_h, A, b)$, by

$$\tilde{N}_i^{\tilde{w}_i}(x) = \begin{pmatrix} A & \tilde{A} \end{pmatrix} \begin{pmatrix} h^{w_h}(x) \\ \tilde{h}^{w_{\tilde{h}}}(x) \end{pmatrix} + b \tag{6}$$

with an arbitrary function $\tilde{h}^{w_{\tilde{h}}}(x)$. The new parameters are $\tilde{w}_i = (w_i, w_{\tilde{h}}, \tilde{A})$. Again, Equation (3) can trivially be satisfied by setting $\tilde{A} = 0$. We can think of two modifications of a neural network that can be expressed by this morphism: firstly, a layer can be widened (i.e., increasing the number of units in a fully connected layer or the number of channels in a CNN - the Net2WiderNet transformation of Chen et al. (2015)). Let $h(x)$ be the layer to be widened. For example, we can then set $\tilde{h} = h$ to simply double the width. Secondly, skip-connections by concatenation as used by Huang et al. (2016) can also be expressed. If $h(x)$ itself is a sequence of layers, $h(x) = h_n(x) \circ \cdots \circ h_0(x)$, then one could choose $\tilde{h}(x) = x$ to realize a skip from $h_0$ to the layer subsequent to $h_n$.

**Network morphism Type III.** By definition, every idempotent function $N_i^{w_i}$ can simply be replaced by

$$N_i^{(w_i, \tilde{w}_i)} = N_i^{\tilde{w}_i} \circ N_i^{w_i} \tag{7}$$

with the initialization $\tilde{w}_i = w_i$. This trivially also holds for idempotent functions without weights, e.g., ReLU.

**Network morphism Type IV.** Every layer $N_i^{w_i}$ is replaceable by

$$\tilde{N}_i^{\tilde{w}_i}(x) = \lambda N_i^{w_i}(x) + (1 - \lambda)h^{w_h}(x), \quad \tilde{w}_i = (w_i, \lambda, w_h) \tag{8}$$

with an arbitrary function $h$ and Equation (3) holds if the learnable parameter $\lambda$ is initialized as 1. This morphism can be used to incorporate any function, especially any non-linearity. For example, Wei et al. (2016) use a special case of this operator to deal with non-linear, non-idempotent activation functions. Another example would be the insertion of an additive skip connection, which were proposed by He et al. (2016) to simplify training: If $N_i^{w_i}$ itself is a sequence of layers, $N_i^{w_i} = N_{i_n}^{w_{i_n}} \circ \cdots \circ N_{i_0}^{w_{i_0}}$, then one could choose $h(x) = x$ to realize a skip from $N_{i_0}^{w_{i_0}}$ to the layer subsequent to $N_{i_n}^{w_{i_n}}$.

Note that every combination of network morphisms again yields a network morphism. Hence, one could, for example, add a block "Conv-BatchNorm-ReLU" subsequent to a ReLU layer by using Equations (4), (5) and (7).

### A.1.2 DETAILS ON APPROXIMATE NETWORK MORPHISMS

Let $T$ be an operator on some space of neural networks $\mathcal{N}(\mathcal{X})$, $p(x)$ a distribution on $\mathcal{X}$ and $\epsilon > 0$. We say $T$ is an $\epsilon$-approximate network morphism (ANM) with respect to a neural network $N^w$ with parameters $w$ iff

$$\Delta(T, N, w) := \min_{\tilde{w}} E_{p(x)}\big[d\big(N^w(x), (TN)^{\tilde{w}}(x)\big)\big] \leq \epsilon, \tag{9}$$

for some measure of distance $d$. Obviously, every network morphism is an $\epsilon$-approximate network morphism (for every $\epsilon$) and the optimal $\tilde{w}$ is determined by the function-preserving property.

Unfortunately, one will not be able to evaluate the right hand side of Equation (9) in general since the true data distribution $p(x)$ is unknown. Therefore, in practice, we resort to its empirical counterpart $\tilde{\Delta}(T, N, w) := \min_{\tilde{w}} \frac{1}{|X_{train}|} \sum_{x \in X_{train}} d\big(N^w(x), (TN)^{\tilde{w}}(x)\big)$ for given training data $X_{train} \subset \mathcal{X}$. An approximation to the optimal $\tilde{w}$ can be found with the same algorithm as for training, e.g., SGD. This approach is akin to knowledge distillation (Hinton et al., 2015). We simply use categorical crossentropy as a measure of distance.

As retraining the entire network via distillation after applying an ANM is still very expensive, we further reduce computational costs as follows: in cases where the operators only affect some layers in the network, e.g., the layer to be removed as well as its immediate predecessor and successor layers, we let $TN$ first inherit all weights of $N$ except the weights of the affected layers. We then freeze the weights of unaffected layers and train only the affected weights for a few epochs.

In our experiments, we employ the following ANM's: (i) remove a randomly chosen layer or a skip connection, (ii) prune a randomly chosen convolutional layer (i.e., remove $1/2$ or $1/4$ of its filters), and (iii) substitute a randomly chosen convolution by a depthwise separable convolution. We train the affected layers for 5 epochs as described above to minimize the left hand side of Equation 9. Note that these operators could easily be extended by sophisticated methods for compressing neural networks (Han et al., 2016; Cheng et al., 2018).

## A.2 ADDITIONAL EXPERIMENTS

We ran another experiment on CIFAR-10 to compare against additional baselines and also conducted some ablation studies. For the sake of simplicity and computational resource, we solely optimize two objectives: beside validation error as first objective to be minimized, we use $\log(\#params(N))$ as a second objective as a proxy for memory consumption. We again use an identical training setup to guarantee that differences in performance are actually due to the model architecture.

### A.2.1 ADDITIONAL BASELINES

We compare the performance of the following methods and hand-crafted architectures:

1. `LEMONADE` on Search Space I (i.e., searching for entire architectures)

2. `LEMONADE` on Search Space II (i.e., seachring for convolutional cells)

3. Networks from generation 1 of `LEMONADE`. One could argue that the progress in Figure 2 is mostly due to pretrained models being trained further. To show that this is not the case, we also evaluated all models from generation 1.

4. Different-sized versions of MobileNet V1,V2 (Howard et al., 2017; Sandler et al., 2018); these are manually designed architecture aiming for small resource-consumption while retaining high predictive performance.

5. Different-sized NASNets (Zoph et al., 2018); NASNets are the result of neural architecture search by reinforcement learning and previously achieved state-of-the-art performance on CIFAR-10.

6. A random search baseline, where we generated random networks, trained and evaluated them and computed the resulting Pareto front (with respect to the validation data). The number and parameter range of these random networks as well as the training time (for evaluating validation performance) was exactly the same as for `LEMONADE` to guarantee a fair comparison.

Results are illustrated in Figure 5; we also refer to Table 2 for detailed numbers.

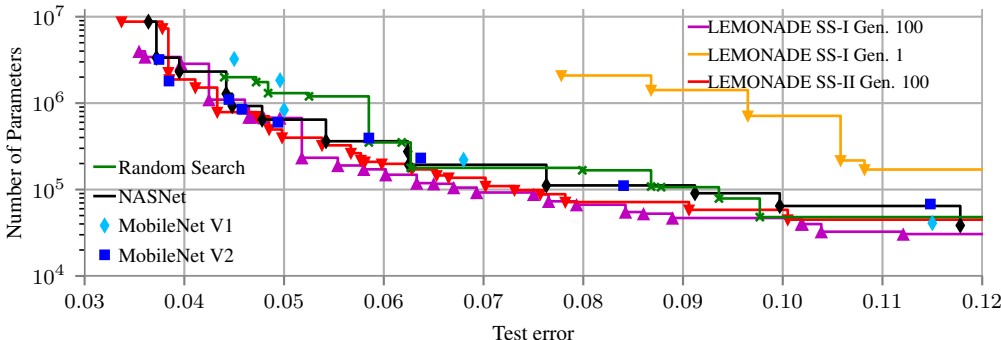

Figure 5: Performance on CIFAR-10 test data of models that have been trained under identical conditions.

| MODEL | PARAMS | ERROR (%) |
|---|---|---|
| MOBILENET | 40K | 11.5 |
| MOBILENET V2 | 68K | 11.5 |
| NASNET | 38K | 12.0 |
| RANDOM SEARCH | 48K | 10.0 |
| LEMONADE | 47K | **8.9** |
| MOBILENET | 221K | 6.8 |
| MOBILENET V2 | 232K | 6.3 |
| NASNET | 193K | 6.8 |
| RANDOM SEARCH | 180K | 6.3 |
| LEMONADE | 190K | **5.5** |
| MOBILENET | 834K | 5.0 |
| MOBILENET V2 | 850K | **4.6** |
| NASNET | 926K | 4.7 |
| RANDOM SEARCH | 1.2M | 5.3 |
| LEMONADE | 882K | **4.6** |
| MOBILENET | 3.2M | 4.5 |
| MOBILENET V2 | 3.2M | 3.8 |
| NASNET | 3.3M | 3.7 |
| RANDOM SEARCH | 2.0M | 4.4 |
| LEMONADE | 3.4M | **3.6** |

Table 2: Comparison between LEMONADE (SS-I), Random Search, NASNet, MobileNet and MobileNet V2 on CIFAR-10 for different model sizes.

### A.2.2  ABLATION STUDY

LEMONADE essentially consists of three components: (i) additionally using approximate network morphism operators to also allow shrinking architectures, (ii) using Lamarckism, i.e., (approximate) network morphisms, to avoid training from scratch, and (iii) the two-staged sampling strategy. In Figure 6, we present results for deactivating each of these components one at a time. The result shows that all three components improve LEMONADE's performance.

### A.3  EXPERIMENTAL DETAILS

In this section we list all the experimental details.

### A.3.1  DETAILS ON SEARCHING FOR ENTIRE ARCHITECTURES (SEARCH SPACE I)

Search Space I corresponds to searching for an entire architecture (rather than cells). LEMONADE's Pareto front was initialized to contain four simple convolutional networks with relatively large validation errors of $30 - 50\%$. All four initial networks had the following structure: three Conv-

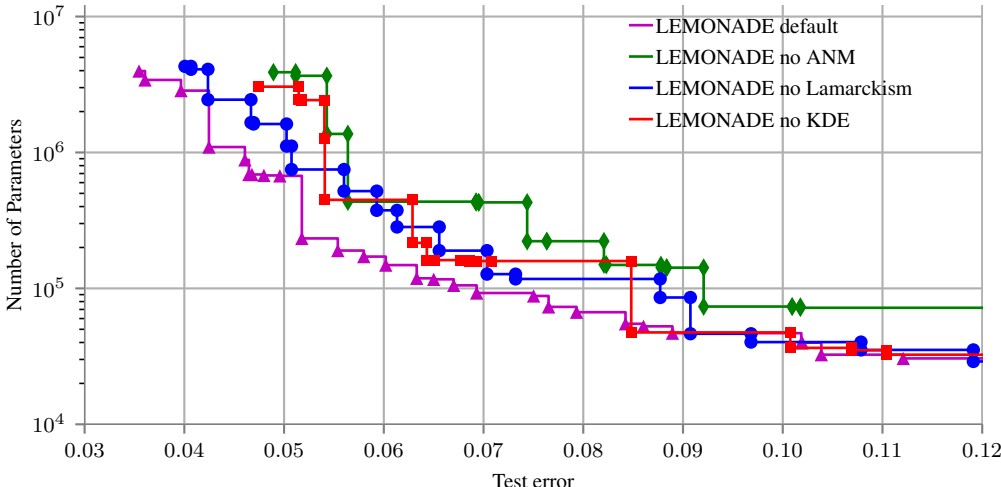

Figure 6: Ablation study on CIFAR-10. We deactivate different components of LEMONADE and investigate the impact. LEMONADE default: Performance of LEMONADE as proposed in this work. LEMONADE no ANM: we deactivated the approximate network morphisms operators, i.e., networks can only grow in size. LEMONADE no Lamarckism: all networks are initialized from scratch instead by means of (approximate) network morphisms. LEMONADE no KDE: we deactivate the proposed sampling strategy and use uniform sampling of parents and children instead.

BatchNorm-ReLU blocks with intermittent Max-Pooling, followed by a global average pooling and a fully-connected layer with softmax activation. The networks differ in the number of channels in the convolutions, and for further diversity two of them used depthwise-separable convolutions. The models had 15 000, 50 000, 100 000 and 400 000 parameters, respectively. For generating children in LEMONADE, we chose the number of operators that are applied to parents uniformly from {1,2,3}.LEMONADE natively handles this unconstrained, arbitrary large search space, whereas other methods are by design restricted a-priori to relatively small search spaces (Bender et al., 2018; Liu et al., 2018b).

We restricted the space of neural architectures such that every architecture must contain at least 3 (depthwise separable) convolutions with a minimum number of filters, which lead to a lower bound on the number of parameters of approximately 10 000.

The network operators implicitly define the search space, we do not limit the size of discovered architectures.

### A.3.2 Details on searching for convolutional cells (Search Space II)

Search Space II consists of convolutional cells that are used within some macro architecture to build the neural network. In the experiments in Section 5, we use cells within the macro architecture of the Shake-Shake architecture (Gastaldi, 2017), whereas in the baseline experiment in the appendix (Section A.2), we rely on a simpler scheme as in as in Liu et al. (2017), i.e., sequentially stacking cells. We only choose a single operator to generate children, but the operator is applied to all occurrences of the cell in the architecture. The Pareto Front was again initialized with four trivial cells: the first two cells consist of a single convolutional layer (followed by BatchNorm and ReLU) with $F = 128$ and $F = 256$ filters in the last block, respectively. The other two cells consist of a single depthwise separable convolution (followed by BatchNorm and ReLU), again with either $F = 128$ or $F = 256$ filters.

### A.3.3 Details on varying the size of MobileNets and NASNets

To classify CIFAR-10 with MobileNets V1 and V2, we replaced three blocks with stride 2 with identical blocks with stride 1 to adapt the networks to the lower spatial resolution of the input.

We chose the replaced blocks so that there are the same number of stride 1 blocks between all stride 2 blocks. We varied the size of MobileNets V1 and V2 by varying the width multiplier $\alpha \in \{0.1, 0.2, \ldots, 1.2\}$ and NASNets by varying the number of cell per block ($\in \{2, 4, 6, 8\}$) and number of filters ($\in \{96, 192, 384, 768, 1536\}$) in the last block.

### A.3.4   DETAILS ON CIFAR-10 TRAINING

We apply the standard data augmentation scheme described by Loshchilov & Hutter (2017), as well as the recently proposed methods mixup (Zhang et al., 2017) and Cutout (Devries & Taylor, 2017). The training set is split up in a training (45.000) and a validation (5.000) set for the purpose of architecture search. We use weight decay ($5 \cdot 10^{-4}$) for all models. We use batch size 64 throughout all experiments. During architecture search as well as for generating the random search baseline, all models are trained for 20 epochs using SGD with cosine annealing (Loshchilov & Hutter, 2017), decaying the learning rate from $0.01$ to $0$. For evaluating the test performance, all models are trained from scratch on the training and validation set with the same setup as described above except for 1) we train for 600 epochs and 2) the initial learning rate is set to $0.025$. While searching for convolutional cells on CIFAR-10, LEMONADE ran for approximately 56 GPU days. However, there were no significant changes in the Pareto front after approximately 24 GPU days. The training setup (both during architecture search and final evaluation) is exactly the same as before.

### A.3.5   DETAILS ON IMAGENET64X64 TRAINING

The training setup on ImageNet64x64 is identical to Chrabaszcz et al. (2017).

### A.4   ADDITIONAL FIGURES

Below we list some additional figures.

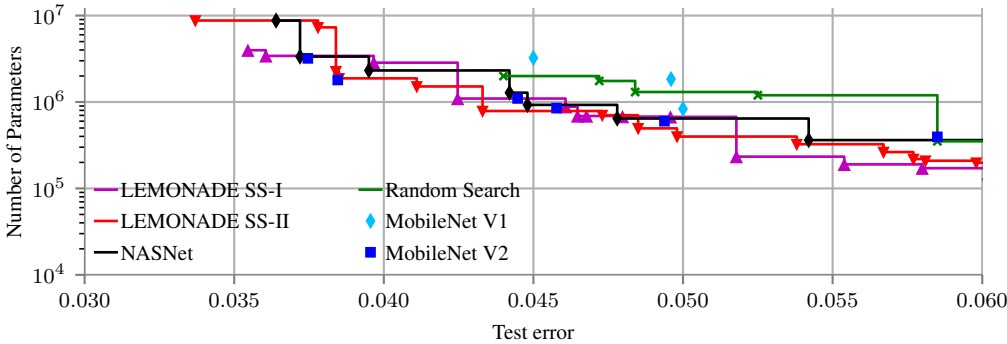

Figure 7: Comparison of LEMONADE with other NAS methods and hand-crafted architectures on CIFAR-10. This plot shows the same results as Figure 5 but zoomed into the range of errors less than 0.06%.

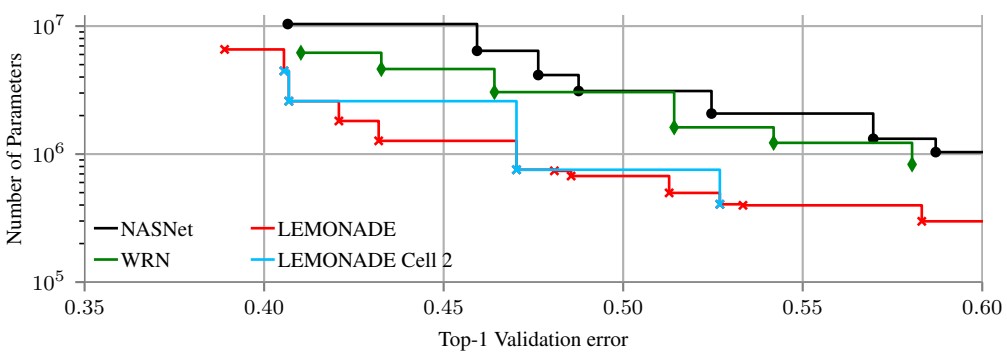

Figure 8: Transferring the cells discovered on CIFAR-10 to ImageNet64x64. Top-1 validation error.

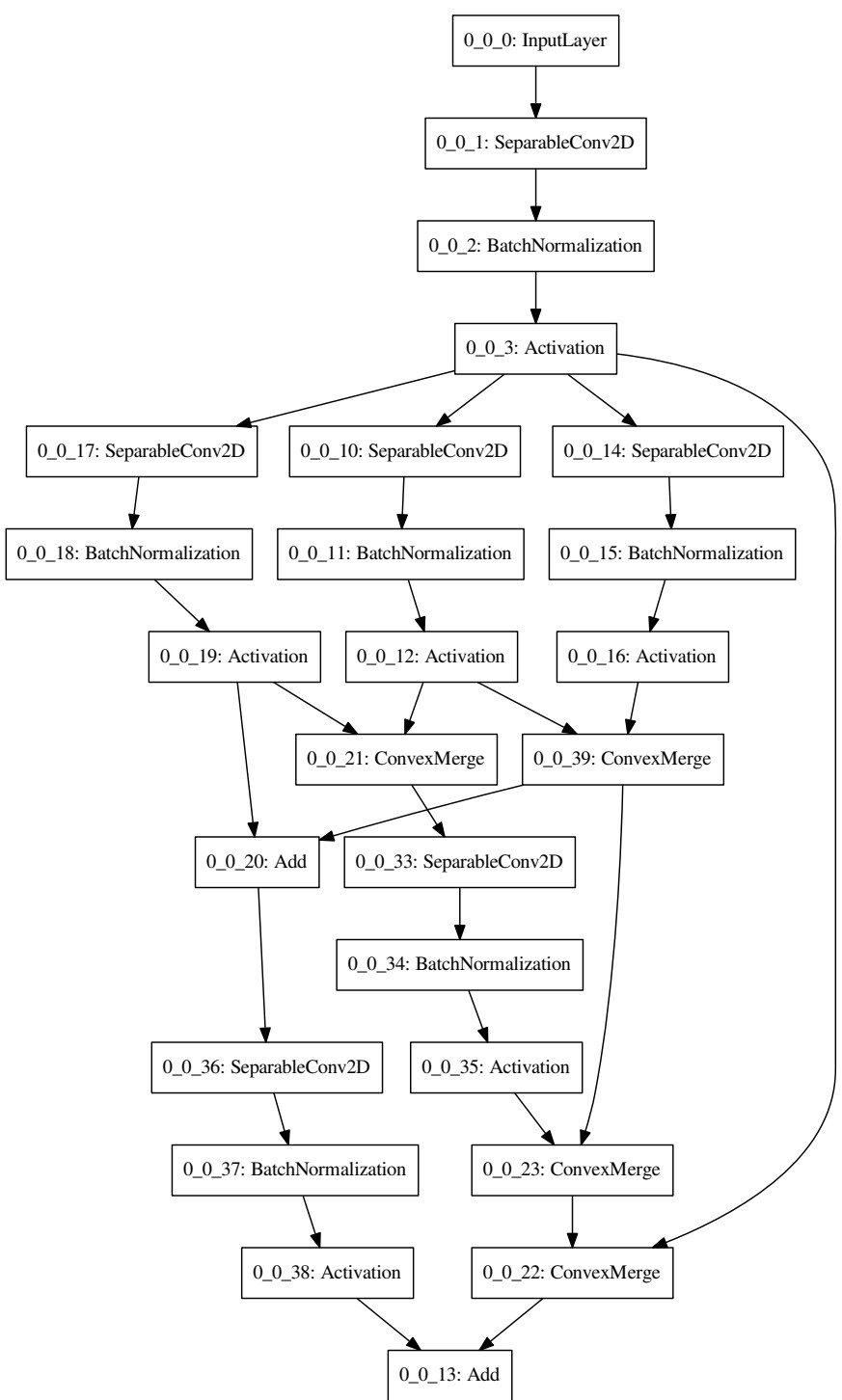

Figure 9: Cell 0. Largest discovered cell.

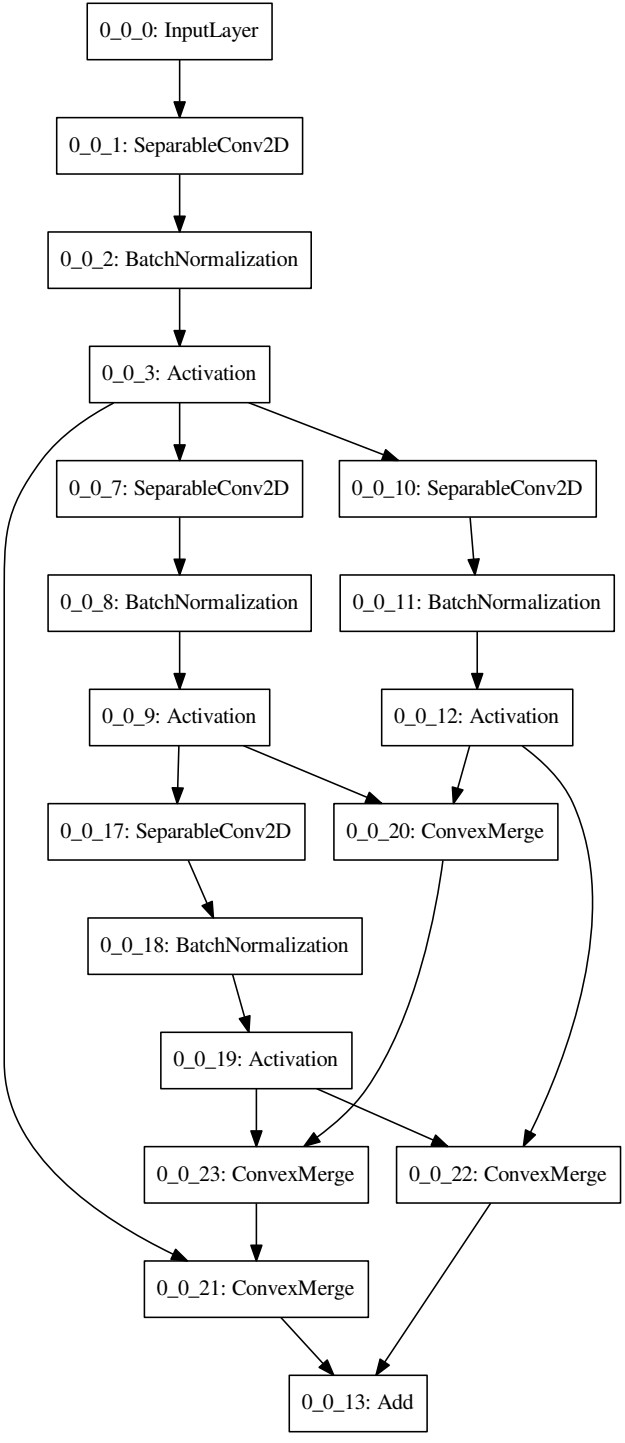

Figure 10: Cell 2

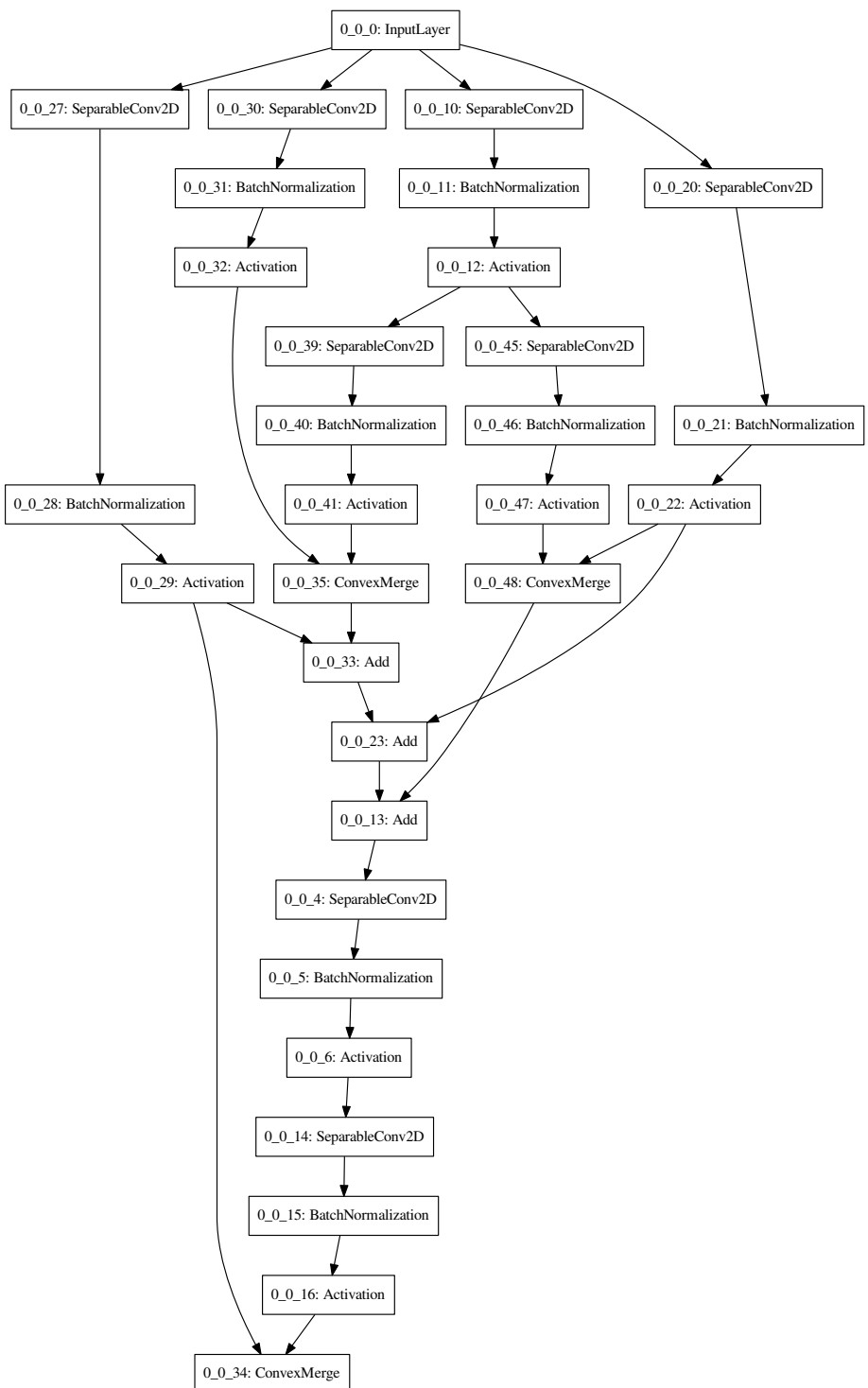

Figure 11: Cell 6

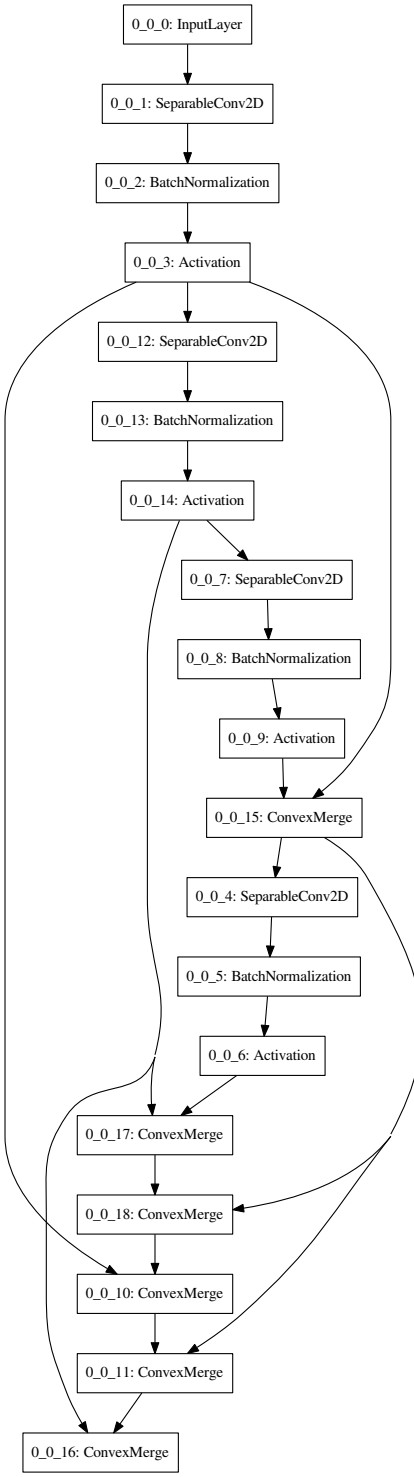

Figure 12: Cell 9

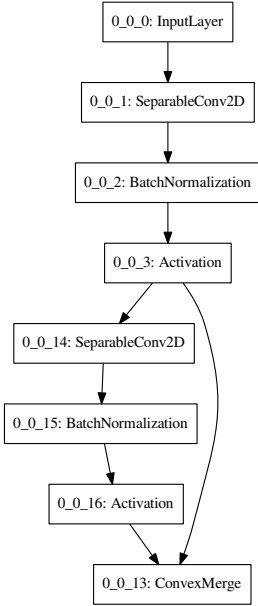

Figure 13: Cell 18

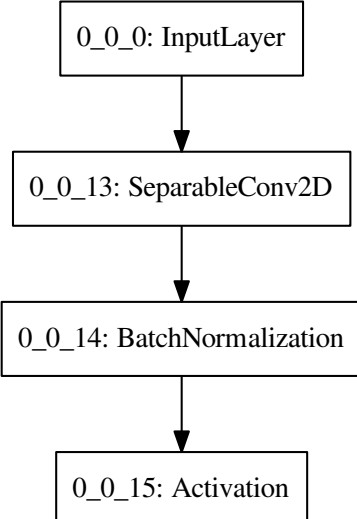

Figure 14: Cell 21. The smallest possible cell in our search space is also discovered.

