# OpenReview forum: "Efficient Multi-Objective Neural Architecture Search via Lamarckian Evolution"
_ICLR.cc/2019/Conference_

### Official Review · AnonReviewer2 · 2018-11-02
**The proposed method is interesting, but the proposed method does not seem to provide a large contribution**

**Rating:** 6
**Confidence:** 3

**Review:**


- Summary
This paper proposes a multi-objective evolutionary algorithm for the neural architecture search. Specifically, this paper employs a Lamarckian inheritance mechanism based on network morphism operations for speeding up the architecture search. The proposed method is evaluated on CIFAR-10 and ImageNet (64*64) datasets and compared with recent neural architecture search methods. In this paper, the proposed method aims at solving the multi-objective problem: validation error rate as a first objective and the number of parameters in a network as a second objective.

- Pros
  - The proposed method does not require to be initialized with well-performing architectures.
  - This paper proposes the approximate network morphisms to reduce the capacity of a network (e.g., removing a layer), which is reasonable property to control the size of a network for multi-objective problems.

- Cons
  - Judging from Table 1, the proposed method does not seem to provide a large contribution. For example, while the proposed method introduced the regularization about the number of parameters to the optimization, NASNet V2 and ENAS outperform the proposed method in terms of the accuracy and the number of parameters.
  - It would be better to provide the details of the procedure of the proposed method (e.g., Algorithm 1 and each processing of Algorithm 1) in the paper, not in the Appendix.
  - In the case of the search space II, how many GPU days does the proposed method require?
  - About line 10 in Algorithm 1, how does the proposed method update the population P? Please elaborate on this procedure.

---

> ### Author Response · Authors · 2018-11-22
> **Answering your questions**
>
> Dear AnonReviewer2,
> thank you for your constructive feedback. Below we address your concerns and questions.
>
> “Judging from Table 1, the proposed method does not seem to provide a large contribution. For example, while the proposed method introduced the regularization about the number of parameters to the optimization, NASNet V2 and ENAS outperform the proposed method in terms of the accuracy and the number of parameters.“
> → The authors of NASNet only provide results for two regimes of parameters (3.3M and  27M) as they do not perform multi-objective optimization but rather just vary two parameters for building NASNet models (number of cells stacked, number of filters). Their method might be optimized to yield good results in these regimes and, admittedly, LEMONADE does not outperform NASNet for models with ~4M parameters. However, from Figure 3 and Table 2 one can see that only varying these two parameters for NASNet models is not necessarily sufficient to generate good models across all parameter regimes. E.g., LEMONADE clearly outperforms NASNet for very small models (50k params, 200k params - Table 2). We also refer to Appendix 3 (“LEMONADE with 5 objectives”), Figure 6, in the updated version of our paper, where one can see that while NASNet has quite strong performance in terms of error, number of parameters and number of multiply-add operations, it performs poorly in terms of inference time. Hence, there is a benefit in doing multi-objective optimization if one is actually interested in multiple objectives and diverse models rather than a single model. This is the main contribution of our paper and different to, e.g., the NASNet paper. The same likely also applies for ENAS (as they use the same search space and conduct very similar experiments).  We also would like to highlight two things: 1) NASNet requires 40x computational resources than LEMONADE, so even if NASNet performs better for ~4M parameter models, LEMONADE achieves competitive performance in significantly less time. 2) Table 1 shows results for models trained with different training pipelines and hyperparameters, and hence it is hard to say architecture X performs better than architecture Y since differences could simply be due to e.g. different learning rates, batch sizes, etc.  In contrast, all other results in the paper (e.g., Figure 3 and Table 2) provide comparisons with exactly the same training pipeline and hyperparameters. .
>
> “It would be better to provide the details of the procedure of the proposed method (e.g., Algorithm 1 and each processing of Algorithm 1) in the paper, not in the Appendix. “
> -> Thanks, we agree; we re-organized our paper accordingly.
>
>
> “- In the case of the search space II, how many GPU days does the proposed method require?
> -> We also ran this experiments for 7*8 GPU days, however the method converged after roughly 3*8 GPU days (meaning that there were no significant differences afterwards).
>
> “About line 10 in Algorithm 1, how does the proposed method update the population P? Please elaborate on this procedure.”
> -> The population is updated to be all non-dominated points from the current population and the generated children, i.e. the Pareto frontier based on all current models. We clarified this in Algorithm 1. Thanks for pointing us towards this.
>
>
> We hope this clarifies your questions. Thanks again for the review!

---

### Official Review · AnonReviewer1 · 2018-11-03

**Rating:** 6
**Confidence:** 4

**Review:**

Summary:
The paper proposes LEMONADE, an evolutionary-based algorithm the searches for neural network architectures under multiple constraints. I will say it first that experiments in the paper only actually address to constraints, namely: log(#params) and (accuracy on CIFAR-10), and the method as currently presented does not show possible generalization beyond these two objectives, which is a weakness of the paper.

Anyhow, for the sake of summary, let’s say the method can actually address multiple, i.e. more than 2, objectives. The method works as follows.

1. Start with an architecture.

2. Apply network morphisms, i.e. operators that change a network’s architecture but also select some weights that do not strongly alter the function that the network represents. Which operations to apply are sampled according to log(#params). Details are in the paper.

3. From those sampled networks, the good ones are kept, and the evolutionary process is repeated.

The authors propose to use operations such as “Net2WiderNet” and “Net2DeeperNet” from Chen et al (2015), which enlarge the network but also choose a set of appropriate weights that do not alter the function represented by the network. The authors also propose operations that reduce the network’s size, whilst only slightly change the function that the network represented.

Experiments in the paper show that LEMONADE finds architecture that are Pareto-optimal compared to existing model. While this seems like a big claim, in the context of this paper, this claim means that the networks found by LEMONADE are not both slower and more wrong than existing networks, hand-crafted or automatically designed.

Strengths:
1. The method solves a real and important problem: efficiently search for neural networks that satisfy multiple properties.

2. Pareto optimality is a good indicator of whether a proposed algorithm works on this domain, and the experiments in the paper demonstrate that this is the case.

Weaknesses:
1. How would LEMONADE handle situations when there are more than one $f_{cheap}$, especially when different $f_{cheap}$ may have different value ranges? Eqn (8) and Eqn (9) does not seem to handle these cases.

2. Same question with $f_{exp}$. In the paper the only $f_{exp}$ refers to the networks’ accuracy on CIFAR-10. What happens if there are multiple objectives, such as (accuracy on CIFAR-10, accuracy on ImageNet) or (accuracy on CIFAR-10, accuracy on Flowers, image segmentation on VOC), etc.

I thus think the “Multi-Objective” is a bit overclaimed, and I strongly recommend that the authors adjust their claim to be more specific to what their method is doing.

3. What value of $\epsilon$ in Eqn (1) is used? Frankly, I think that if the authors train their newly generated children networks using some gradient descent methods (SGD, Momentum, Adam, etc.), then how can they guarantee the \epsilon-ANM condition? Can you clarify and/or change the presentation regarding to this part?

---

> ### Author Response · Authors · 2018-11-22
> **Answering the questions**
>
> Dear AnonReviewer1,
> thank you for your positive and constructive feedback. Below we address your concerns and questions.
>
> “What value of $\epsilon$ in Eqn (1) is used? [...] how can they guarantee the \epsilon-ANM condition?”
> → Indeed, one can not guarantee the \epsilon-ANM condition for an arbitrary epsilon. However, in our application one does not need to explicitly select $\epsilon$ at all. We simply apply an approximate network morphism operator. Case 1, epsilon is small: the output is a network that is “smaller” than its parent and has a similar error, so the children will likely be non-dominated and it will be part of the pareto front in the next generation. Case 2, epsilon is large (hence likely also the error): the children will likely be dominated by some other network and it will be discarded when the Pareto front is updated. Thus, in both cases, the specific epsilon doesn’t matter. The step of LEMONADE, where the Pareto front is updated, will automatically decide whether the morphing was successful or not based on the (non-)domination criterion. We updated (shortened) the section on approximate network morphism to not put a too strong emphasis on this. Hopefully it is now less confusing.
>
>
> “[...] the method as currently presented does not show possible generalization beyond these two objectives, which is a weakness of the paper.”
> -> We respectfully disagree. In principle, the proposed method is - as is - applicable to arbitrary objectives and arbitrary many objectives. It is neither restricted to these specific objectives nor to n=2 objectives. To demonstrate this, we carried out a new experiment with exactly the same method on 5 objectives (2 expensive ones, 3 cheap ones). We refer to the additional experiment, Appendix 3 (“LEMONADE with 5 objectives”), in the updated version of our paper.
>
> “How would LEMONADE handle situations when there are more than one $f_{cheap}$, especially when different $f_{cheap}$ may have different value ranges? Eqn (8) and Eqn (9) does not seem to handle these cases.”
> -> Both equations are not restricted to 1D inputs. (Kernel) density estimators can, in general,  be applied to arbitrary dimensions and most packages allow multi-dimensional inputs by default (e.g. KDE in scipy or scikit-learn). Of course, density estimation becomes problematic with increasing number of dimensions, but we believe 4-6 objectives is a realistic dimensionality for NAS applications, and scaling to significantly more objectives will typically not be necessary.
>
> Note that the output of a KDE is always 1D, independent of the input. Also, most packages provide methods for, e.g., automatic bandwidth computation (per input dimension) to handle different value ranges. To make the input and output spaces in equations 8,9 (equations 1,2 in the updated version) clearer, we provide them in detail here:
> f_cheap: <some neural network space>  → R^n, where n is the number of cheap objectives
> p_kde: R^n → R
> p_p: <some neural network space> → R
>
> “Same question with $f_{exp}$.”
> → The expensive objectives are only involved in the last two steps of LEMONADE (evaluate $f_{exp}$ on the subset of children, update the Pareto frontier). These steps can be applied to more than one expensive objective. E.g. instead of training the children only on CIFAR-10, we can also train them on some other data set as well (and in our new experiment with 5 objectives we indeed also train them on CIFAR-100 as a second expensive objective). Of course, the runtime of the method will increase linearly in the number of expensive objectives.
>
> So, to summarize regarding having only 2 objectives:
> 1) Our method can in principle handle more than 2 objectives (both cheap and expensive), there is no general restriction to n=2 objectives.
> 2) From an implementation point of view, common packages for computing density estimators automatically deal with multi-dimensional inputs and different ranges, hence LEMONADE can be run  - as is -  with multi-dimensional objectives without any further user interaction or modifications.
> 3) To confirm these statements, we ran an additional experiment with 5 objectives - 2 expensive ones (performances on Cifar-10, performance on Cifar-100) and 3 cheap ones (number of parameters, number of multiply-add operations, inference time). We refer to Appendix 3, “LEMONADE with 5 objectives”, in the updated version of our paper for details and results.
>
>
> We hope this clarifies your questions. Thanks again for the review!

---

### Official Review · AnonReviewer3 · 2018-11-04
**An interesting method with a troubled presentation**

**Rating:** 6
**Confidence:** 3

**Review:**

This paper proposes LEMONADE, a random search procedure for neural network architectures (specifically neural networks, not general hyperparameter optimization) that handles multiple objectives.  Notably, this method is significantly more efficient more efficient than previous works on neural architecture search.

The emphasis in this paper is very strange.  It devotes a lot of space to things that are not important, while glossing over the details of its own core contribution.  For example, Section 3 spends nearly a full page building up to a definition of an epsilon-approximate network morphism, but this definition is never used.  I don't feel like my understanding of the paper would have suffered if all Section 3 had been replaced by its final paragraph.  Meanwhile the actual method used in the paper is hidden in Appendices A.1.1-A.2.   Some of the experiments (eg. comparisons involving ShakeShake and ScheduledDropPath, Section 5.2) could also be moved to the appendix in order to make room for a description of LEMONADE in the main paper.

That said, those complaints are just about presentation and not about the method, which seems quite good once you take the time to dig it out of the appendix.

I am a bit unclear about how comparisons are made to other methods that do not optimize for small numbers of parameters? Do you compare against the lowest error network found by LEMONADE? The closest match in # of parameters?

Why is the second objective log(#params) instead of just #params when the introduction mentions explicitly that tuning the scales between different objectives is not needed in LEMONADE?

It seems like LEMONADE would scale poorly to more than 2 objectives, since it effectively requires approximating an #objectves-1 dimensional surface with the population of parents.  How could scaling be handled?

---

> ### Author Response · Authors · 2018-11-22
> **Answering the questions**
>
> Dear AnonReviewer3,
> thank you for your positive review and constructive feedback!
>
> We agree that the structure of the paper was not optimal and reorganized it along the lines you suggested (thanks for the suggestion!). Below we address specific questions.
>
> “I am a bit unclear about how comparisons are made to other methods that do not optimize for small numbers of parameters? Do you compare against the lowest error network found by LEMONADE? The closest match in # of parameters?”
> -> The latter: we compared with the models with the closest match in # of parameters.
>
> “Why is the second objective log(#params) instead of just #params when the introduction mentions explicitly that tuning the scales between different objectives is not needed in LEMONADE?”
> -> We stated that defining a trade-off between objectives is not necessary (in case you are referring to this statement), which would, e.g., be necessary when one would scalarize objectives by using a weighted sum. Rescaling an objective, however, is different as it is independent from other objectives: it only depends on that specific objective and which scale is important to the user and the application. For the number of parameters, the log scale is natural to cover a large range of sizes: think of a plot of size vs. performance; in order to see anything for small sizes one would typically put the size on a log scale (and we indeed did, see, e.g., Figures 3 and 4). Therefore, it is most natural to also put the number of parameters on a log scale for LEMONADE.
>
> “It seems like LEMONADE would scale poorly to more than 2 objectives, since it effectively requires approximating an #objectives-1 dimensional surface with the population of parents. How could scaling be handled?”
> -> We think having 4-6 objectives is a realistic dimensionality for NAS applications, and scaling to significantly more objectives (which would indeed be problematic for our method, but also for multi-objective optimization in general) is typically not necessary. In response to this question, to demonstrate this, wee conducted a new experiment with 5 objectives (performance on Cifar 10, performance on Cifar 100, number of parameters, number of multiply-add operations, inference time) to show that LEMONADE can handle these realistic scenarios natively. We refer to the updated version of our paper for the results (Appendix 3,“LEMONADE with 5 objectives”), but in a nutshell the results are very positive and qualitatively resemble those for two objectives.
> While we put this experiment into the appendix for now to not change the main paper too much compared to the submitted version, if the reviewers agree we would also be very happy to include this experiment in the main paper.
>
> We hope this clarifies your questions. Thanks again for the review!

---

### Author Response · Authors · 2018-11-22
**Revised version of our paper online!**

Dear reviewers,

thanks again for your valuable feedback. We just updated our paper. We mainly made two modifications, based on your feedback:
1) We reorganized the paper according to your suggestions; some parts of the main paper were moved to the appendix, some parts of the appendix were moved to the main paper.
2)As you were asking whether LEMONADE is applicable to more than 2 objectives, we ran an experiment with 5 objectives, namely 1) performance on Cifar 10 (expensive objective), 2) performance on Cifar 100 (expensive) , 3) number of parameters (cheap), 4) number of multiply-add operations (cheap), 5) inference time (cheap). We refer to Appendix 3, “LEMONADE with 5 objectives”, for details and results, but in a nutshell the results are very positive and qualitatively resemble those for two objectives. While we put this experiment into the appendix for now to not change the main paper too much compared to the submitted version, if you agree we would also be very happy to include this experiment in the main paper.

We hope the updated version and our answers to your reviews have cleared out all major concerns and we kindly ask you to update your rating if we clarified your concerns.

---

### Meta-Review · Area_Chair1 · 2018-12-14
**interesting method, promising results**

**Confidence:** 5
**Recommendation:** Accept (Poster)

**Metareview:**

The paper proposes an evolutionary architecture search method which uses weight inheritance through network morphism to avoid training candidate models from scratch.  The method can optimise multiple objectives (e.g. accuracy and inference time), which is relevant for practical applications, and the results are promising and competitive with the state of the art. All reviewers are generally positive about the paper. Reviewers’ feedback on improving presentation and adding experiments with a larger number of objectives has been addressed in the new revision.

I strongly encourage the authors to add experiments on the full ImageNet dataset (not just 64x64) and/or language modelling -- the two benchmarks widely used in neural architecture search field.